John R. Worden[1], Susan S. Kulawik[2], Dejian Fu[1], Vivienne H. Payne[1], Alan E. Lipton[3], Igor Polonsky[3], Yuguang He[3], Karen Cady-Pereira[3], Jean-Luc Moncet[3], Robert L. Herman[1], Frederick W. Irion[1], and Kevin W. Bowman[1]

1. Jet Propulsion Laboratory / California Institute for Technology, Pasadena, CA

2. Bay Area Environmental Research Institute, Mountain View CA, USA

3. Atmospheric Environmental Research, Lexington MA, USA

**Characterization and Evaluation of AIRS-Based Estimates of the Deuterium Content of Water Vapor**

**Abstract**: Single pixel, tropospheric retrievals of HDO and $H_2O$ concentrations are retrieved from Atmospheric Infrared Sounder (AIRS) radiances using the optimal estimation algorithm developed for the Aura Tropospheric Emission Spectrometer (TES) project. We evaluate the error characteristics and vertical sensitivity of AIRS measurements corresponding to five days of TES data (or 5 global surveys) during the Northern Hemisphere summers between 2006 and 2010 (~600 co-located comparisons per day). We find that the retrieval characteristics of the AIRS deuterium content measurements have similar vertical resolution middle-troposphere as TES but with slightly less sensitivity in the lower-most troposphere, with a typical degrees-of-freedom (DOFS) in the tropics of 1.5 and approximately double the uncertainty. The calculated measurement uncertainty is ~30 per mil (parts per thousand relative to the deuterium composition of ocean water) for a tropospheric average between 750 and 350 hPa, the altitude region where AIRS is most sensitive. Comparison with the TES data also indicate that the uncertainty of a single target AIRS HDO/$H_2O$ measurement is ~30 per mil. Comparison of AIRS and TES data between 30 degrees South and 50 degrees North indicate that the AIRS data is biased low by ~-2.6 per mil with a latitudinal variation of ~7.8 per mil. This latitudinal variation is consistent with the accuracy of TES data as compared to in situ measurements, suggesting that both AIRS and TES have similar accuracy.

**Introduction**:

Measurements of the isotopic composition of water can help identify the source of the water and provide knowledge about its condensation and evaporation history (e.g. Galewsky *et al*. and refs therein). Through most of the twentieth century, most isotopic measurements of water have been of precipitation (e.g. Craig, 1961). Near global measurements of the isotopic composition of water vapor became possible with the advent of spectroscopic techniques applied to in situ measurements (e.g., Noone *et al*., 2011) using lasers and for passive ground based and satellite measurements (e.g. Worden *et al*., 2006; Frankenberg *et al*. 2009; Schneider *et al*. 2012; Lacour *et al*. 2012). These data have in turn been used to evaluate the role of convection, large scale dynamics, and evapotranspiration on the tropical water cycle (e.g. Worden *et al*. 2007; Frankenberg *et al*. 2009; Wright *et al*. 2017) tropical convection (e.g. Lacour 2018 and refs therein) and the role of plants on global evapotranspiration (Good *et al*. 2015).

In this paper we demonstrate a retrieval algorithm, based upon the Aura TES optimal estimation retrieval algorithm (e.g. Worden *et al*. 2012) that can provide robustly characterized measurements of the deuterium content of water vapor (HDO and $H_2O$) from the AIRS measurements. Our goal is to create a multi-decadal Earth Science Data Record (ESDR) using the AIRS and TES data; the TES global record spans ~6 years (2005-2010) and the AIRS data span 17+ years starting in 2002. This ESDR could potentially be used for evaluating the changing water cycle (e.g. Bailey *et al*., 2017) and its coupling to the carbon cycle (e.g. Zhou *et al.,* 2014; Wright *et al*., 2017).

We first characterize the vertical resolution and uncertainties for estimates of HDO and $H_2O$, and their ratio, using AIRS radiance observations corresponding to boreal summertime TES global survey's between 2006 through 2010, which is the time period when TES observations sample the (near) global atmosphere and the calibration approach for TES measurements remained the same. We make only these comparisons due to current processing limitations but expect additional overlap between TES and AIRS data sets in the coming years. We then compare the AIRS and TES data to evaluate the calculated uncertainties of the AIRS data.

1) **Description of AIRS and TES instruments**

The AIRS instrument is a nadir-viewing, scanning infrared spectrometer (Aumann *et al*. 2003; Pagano *et al.,* 2003; Irion *et al*. 2018; DeSouza-Machado *et al*. 2018) that is onboard the

NASA Aqua satellite and was launched in 2002. AIRS measures the thermal radiance between approximately 3-12 microns with a resolving power of approximately 1200. For the 8 micron spectral range used for the HDO/$H_2O$ retrievals, the spectral resolution is ~1 cm$^{-1}$ with a gridding of ~0.5 cm$^{-1}$; the signal-to-noise (SNR) ranges from ~400 to ~1000 over the 8 micron region for a typical tropical scene. A single footprint has a diameter of ~15 km in the nadir; with the ~1650 km swath, the AIRS instrument can measure nearly the whole globe in a single day. The Aqua satellite is part of the "A-Train" that consists of multiple satellites, including TES, in a sun-synchronous orbit at 705 km with an approximately 1:30 pm equator crossing-time.

The Aura TES instrument is a Fourier Transform Spectrometer that originally was designed to measure the thermal infrared (IR) radiances both in the limb and nadir viewing in order to obtain vertically resolved trace gas profiles of ozone, CO, $CH_4$, HDO and $H_2O$, and several ozone pre-cursors such as ammonia, methanol, and PAN (e.g. Beer *et al.*, 2001; Worden *et al.*, 2004; Worden *et al.* 2006; Luo *et al.*, 2007; Beer *et al.* 2008; Worden *et al.*, 2012; Payne *et al.* 2014). Several of these trace gases, such as CO, $CH_4$, and ammonia have also been quantified using AIRS radiances (e.g. McMillan *et al.*, 2005; Xiong *et al.* 2008; Warner *et al.*, 2016). In comparison to the AIRS instrument, TES has a spectral resolution of ~0.12 cm$^{-1}$ (apodized) with a spectral gridding of 0.06 cm$^{-1}$. The SNR is ~300 in the 8 microns spectral region. The Aura TES instrument, after the summer of 2005, observes one nadir scene every 100 km along the orbit path. The effective length of the record is approximately five years, between September 2005 through November 2009, after which instrument degradation problems resulted in interrupts and a decrease in sampling. The AIRS instrument has nearly one thousand times the sampling of TES and near continuous operation between 2002 through the present and therefore can be used to construct several composition based ESDR's.

**3) Description of the Radiative Transfer Forward Model**

The radiative transfer forward model used for this work is the Optimal Spectral Sampling (OSS) fast radiative transfer model (RTM) (Moncet et al., 2015; Moncet et al., 2008). The OSS approach is integrated in the operational Cross-Track Infrared Sounder (CrIS, Han et al. 2013) processing system (Divarkala et al., 2014) and has also been utilized for trace gas retrievals from CrIS (e.g. Shephard and Cady-Pereira, 2015). OSS uses a series of approximations tailored to a

specific frequency range and spectral resolution to increase the radiative transfer calculation
performance by approximately a factor of 20-100 relative to a line-by-line calculation
(http://rtweb.aer.com ). OSS can be trained to user-defined accuracy relative to the line-by-line
model used for training. Here, the training threshold was set to 20 % of the AIRS noise level.
The line-by-line model used as a reference in the training and to build the absorption coefficient
look-up tables (LUTs) used by the fast RTM is the Line-By-Line Radiative Transfer Model
(LBLRTM) (Clough et al., 2005; Alvarado et al., 2013). The OSS version used in this work is
based on LBLRTM v12.4, using the TES_v2.0 spectroscopic line parameter database. The
TES_v2.0 line parameter database follows the HITRAN 2012 compilation (Rothman et al.,
2013]), with the following exceptions:
- $H_2O$ positions and intensities are taken from the aer_v_3.4 line parameter database
(http://rtweb.aer.com), closely following the measured and calculated values published in
Coudert et al. (2008).
- $CH_4$ includes first order line mixing coefficients (as supplied in the aer_v_3.4 line
parameter database). These were calculated using the approach of Tran et al. (2006).
- $CO_2$ line parameters are from the database of Lamouroux et al. (2015). This database
takes most of its line positions, intensities, and lower state energies from the HITRAN
2012 database,  but the values for air-broadening half-widths and their temperature
dependences are adjusted from the HITRAN 2012 values to be consistent
throughout the bands, and the air-induced pressure shifts (not given for a majority of
transitions in HITRAN 2012) were added. The TES_v2.0 database includes first order
line mixing coefficients (as supplied in the aer_v_3.4.1 line parameter database),
calculated using the software of Lamouroux et al. (2015).
Further information on the AER line parameter databases can be found at http://rtweb.aer.com.
OSS is adapted for use with AIRS radiances using the version 4 AIRS spectral response function
(SRF) (Strow et al., 2003) that is interpolated to a uniform grid of 0.004 $cm^{-1}$ centered on the
channel center frequencies. The OSS radiative transfer code provides speedup of 20-100x over
the original TES operational radiation transfer model (Clough *et al*., 2006).

**4) Description of the Retrieval Approach**

2       The optimal estimation algorithm used in this analysis for quantifying $CH_4$, HDO, $H_2O$,

temperature, cloud properties, and emissivity is extensively discussed in Worden et al. (2004),
Bowman et al. (2006),  and Worden et al (2012). We therefore refer the reader to those papers
for a description of the retrieval algorithm, with a suggestion that they start with the Worden *et*
*al*. (2012) paper; however, we will briefly summarize the retrieval approach here. This retrieval
algorithm, now called the MUlti-SpEctra, MUlti-SpEcies,  MUlti-Sensors (MUSES) algorithm
(Worden et al., 2007b; Fu et al., 2013, 2016, 2018; Luo et al., 2013; Worden et al., 2013), can
use radiances from multiple instruments including TES, CrIS, OMI, OMPS, TROPOMI, and
MLS to quantify geophysical observables that affect the corresponding radiance.
For the AIRS retrievals discussed here, we simultaneously estimate not just $CH_4$, CO,
HDO, and $H_2O$ but also temperature (surface and atmosphere), emissivity (if over land), and a
spectrally varying gray body cloud (e.g. Kulawik *et al*., 2006, Eldering *et al.,* 2008). As in
Worden *et al*. (2006) and Worden *et al*. (2012) the constraint matrix used to regularize the HDO
and $H_2O$ components of the retrieval includes off-diagonal components that reflect *a priori*
knowledge about the variability of HDO with respect to $H_2O$ in order to ensure that retrieval of
the ratio of HDO to $H_2O$ is optimized, as opposed to either HDO or $H_2O$ alone.  The prior
information used for this covariance is derived from monthly climatologies using the NCAR
Global Climate Model as discussed in Worden *et al*. (2006).  The *a priori*  profile used for the
HDO/$H_2O$ ratio is set to be constant over the whole globe, and represents the mean tropical *a*
*priori*  profile from the NCAR model. However,  the $H_2O$ *a priori*  profile is allowed to vary by
latitude and is based on re-analysis (Worden *et al*. 2006); therefore the HDO profile is the mean
tropical profile of the HDO/$H_2O$ ratio from the NCAR model multiplied by the $H_2O$ *a priori*
profile.
We use single pixel radiances that have not been transformed through "cloud clearing" in
order to preserve the original, well characterized radiance noise characteristics for use in our
estimates (Irion *et al*. 2018; DeSouza-Machado *et al*. 2018) and because we find that single-pixel
AIRS radiances have sufficient information about cloud pressure and optical depth to be
retrieved jointly with the trace gases, as demonstrated empirically through validation of these
AIRS-based composition retrievals with TES retrievals (e.g. Figures 1-4).  We assume the noise
in any given pixel is uncorrelated with those from adjacent pixels. However, these correlations
are known to exist (e.g. Pagano *et al*. 2008) and the impact of ignoring them is that our
calculated uncertainties will be larger than expected and therefore our noise related uncertainty
should be considered a conservative estimate.
A primary difference between the retrieval approach shown in this paper versus the TES
methane and HDO retrievals (Worden *et al.,* 2012) and those from previous efforts using AIRS
radiances (e.g. Xiong *et al.,* 2008) is that we retrieve these trace gas profiles using the AIRS
radiances from ~8 and ~12 microns instead of radiances from the 8 micron region alone in order
to provide a stronger constraint on atmospheric temperature and hence reduce uncertainty from
knowledge of temperature on the HDO and $H_2O$ retrieval. The 8 micron region used (~1217 to
1315 cm$^{-1}$) for these retrievals has the most sensitivity to HDO and $H_2O$ whereas the 12 micron
band (~650 to 900 cm$^{-1}$) is primarily sensitive to temperature and $H_2O$. All channels are used
within this spectra unless flagged as poor during calibration.
**5) Characterization of HDO/$H_2O$ profiles**
While $H_2O$ is quantified using radiances from both the 12 micron and 8 micron spectral
regions, the primary absorption lines used here to quantify HDO are in the 8 micron region.
There are other HDO (and $H_2O$) lines available to use from the AIRS radiance but for now we
only use the 8 micron region to ensure consistency between AIRS and TES data. Figure 1 shows
the 8 micron radiance (top panel) and the Jacobian, or sensitivity of the radiance to variations in
the (log) $H_2O$ and (log) HDO respectively (middle and bottom panels). These Jacobians are
normalized by the instrument noise. For example, a value of 1 means that it would take a 100%
change in the corresponding species to distinguish between two similar radiances (everything
about the observed scene and radiance is the same except for the species of interest) above the
noise level. A value of ~-50 therefore means that only a 2% variation is required (or 1/50).
Figure 2 shows the averaging kernel matrix for the HDO component of the joint retrieval.
The averaging kernel describes the response of the estimate, or log(HDO), relative to variations
in the true state; consequently it can also be used to evaluate the vertical resolution and
sensitivity of the estimate. For example, if HDO varies by 100% at 908 hPa, then the AIRS
estimate would be able to observe about 30% of the variability because the averaging kernel is
approximately 0.3 at that level. The averaging kernel at 908 hPa also depends on the deuterium
content at several other pressure levels below and above indicating that the estimate at 908 hPa
depends on the deuterium content variations at these other levels. Not shown are the
dependencies of the (log) HDO estimate to those from the (log) $H_2O$ estimate. These
dependencies between the HDO averaging kernels and with the $H_2O$ averaging kernels are
accounted for when constructing the HDO/$H_2O$ ratio; however a residual uncertainty called the
"smoothing" error is imparted when comparing the HDO/$H_2O$ ratio to independent data; this
smoothing error is part of the error budget shown in Figure 3. As discussed in Worden *et al.*
(2012) and Schneider *et al.* (2012), the sensitivity of the estimated HDO/$H_2O$ ratio is limited by
the sensitivity of the estimate to HDO. Users of these data should note that this ratio is typically
used with that of $H_2O$ in order to better evaluate their joint variation (HDO/$H_2O$, $H_2O$) against
simple mixing and rainfall models (Noone *et al.* 2011). However, the sensitivity of the radiance
to $H_2O$ variations is much stronger than that for HDO, although the altitude region of the HDO
sensitivity typically overlaps with the $H_2O$ sensitivity. Schneider *et al.* (2012) discusses how to
created HDO/$H_2O$, $H_2O$ pairs to mitigate this component of the smoothing error when comparing
these data against the simple models described in Noone *et al.* (2011). For comparison to more
complex global climate models the user of these data also needs to apply the HDO and $H_2O$
averaging kernels to the corresponding model fields (e.g. Risi *et al.*, 2012).

18          Figure 3 (top panel) shows the tropospheric deuterium content (or HDO/$H_2O$ ratio)

derived from AIRS observations on July 1 2006. Despite the improved computational
performance of the OSS radiative transfer calculation relative to the TES algorithm line-by-line
calculation (Clough *et al.* 2005), the retrieval is still sufficiently expensive such that we can only
process a sub-set of the AIRS retrievals. Considering the computational cost, for the purpose of
constructing a record we currently only process AIRS retrievals  from between 45 degrees South
to 65 degrees North that coincide with the nearest TES observation but with an additional two
observations within 100 km of the TES track over the continents; this ad hoc sampling strategy is
based on experience with previous studies using the TES deuterium and methane measurements.
The traditional notation for this quantity is called "delta-D" , or "δ-D" with units of "per mil" or
parts per thousand relative to the Standard Mean Ocean Water (SMOW) deuterium content
which is $3.11 \times 10^4$ molecules of HDO per molecule of $H_2O$.  The observations shown represent
the deuterium content for the pressures between 750 hPa and 350 hPa, where we find the AIRS
and TES observations have maximal overlap in their vertical resolution.
The errors are calculated during the optimal estimation retrieval (Bowman *et al*. 2007;
Worden *et al*. 2012) and depend on the expected noise of the AIRS radiances and the parameters
that are co-retrieved with the AIRS HDO/$H_2O$ ratio such as temperature, surface emissivity,
clouds, and methane. As noted in Worden *et al*. (2012) these co-retrieved parameters affect both
the precision and accuracy whereas the noise only affects the precision. The total error (middle
panel) is given in units of per mil and ranges between 25 to 30 per mil. The DOFS, or trace of
the averaging kernel,  are shown in the bottom panel and indicate that many of the HDO/$H_2O$
retrievals can resolve different parts of the troposphere, at least in the tropics, because (as
demonstrated in Figure 2) the rows of the averaging kernels are separated between the boundary
layer region (surface to ~750 hPa) and the free-troposphere (~600 to 300 hPa). However, these
observations cannot completely resolve the total variability in these two regions of the
atmosphere because the total DOFS is typically 1.5 or less and for the measurement to be able to
resolve the variability (to within the calculated error) of the two regions there would need to be
at least 2 DOFS.
**6) Comparison of AIRS and TES HDO/$H_2O$ retrievals**

18       Figure 4 shows a comparison between overlapping AIRS and TES estimates of the

HDO/$H_2O$ ratio for June 1 2006.  The AIRS and TES measurements effectively overlap in space
and within a few seconds in time as the instruments are in the same orbit. However not all the
comparisons shown in Figure 4 overlap as retrievals may be rejected due to poor quality. We
therefore compare all data that are within 200 km in the free troposphere. We do not expect
substantive error to occur due to spatial mismatch of 2 degrees or less because air parcels in the
free-troposphere have length scales that are several hundred kilometers long (e.g. Worden *et al*.
2013). The average between approximately 750 hPa and 350 hPa are shown for when the DOFS
are larger than one for this altitude region. There is a slight bias of -2.7 +/- 1.5 per mil between
TES and AIRS as shown in the top panel.  The calculated and actual (RMS difference between
AIRS and TES) uncertainties are shown and are approximately 30 per mil, primarily driven by
the uncertainty in the AIRS based estimates as the TES based estimates have an uncertainty of
approximately 15 per mil.  Figure 5 shows a direct comparison of the AIRS and TES data. The
correlation is about 0.89 and the one-to-one line (solid line) overlaps this distribution. However

the lowest values likely diverge from the one-to-one line, possibly because the vertical

distribution in the sensitivity depends on the amount of HDO and hence we should expect

differences between the TES and AIRS deuterium measurements for these lower-sensitivity

retrievals.

A comparison of the AIRS and TES HDO/$H_2O$ ratio for five single global surveys taken

between 2006 and 2010 (one global survey per year during boreal summer) is shown in Table 1

and indicates that the overall bias varies between -2.7 to 3.7 per mil. Using all 5 TES global

surveys that are summarized in Table 1 we can construct how AIRS and TES compare as a

function of latitude as shown in Figure 6. Figure 6 is constructed by averaging the difference

between TES and AIRS observations within 5 degree latitudinal bins. The mean bias across

latitudes is ~-2.6 per mil. The error bars shown on the difference is the error on the mean, which

is the Root-Mean-Square (RMS) of the differences divided by the square root of the number of

co-located observations; as this error bar is a measure of precision for each latitude bin, this

comparison demonstrates that there are variations in the comparison that are larger than the

precision and are therefore related to systematic errors in either the TES data or AIRS data or

both.  Variations in these systematic errors can be seen in the latitudinal variability, which has an

RMS variation of ~7.8 per mil for the different latitude bins but can vary by as much as ~-15 to

~+15 per mil in the tropics. Typically these variations are due to a combination of uncertainties

in the spectroscopy along with temperature, water vapor, and surface properties; they may also

be due to "smoothing error" which is related to how differences in the vertical resolution affect

the tropospheric average of the deuterium content shown in these figures (e.g. Worden *et al*.

2004).  This 7.8 per mil variation across latitudes is about the same as the reported accuracy of

the Aura TES delta-d observations that are based on comparisons of TES data with surface and

aircraft measurements (Worden *et al*. 2011; Herman *et al*. 2014).  We therefore report the current

accuracy of the AIRS data to be ~7.8 per mil.  We expect future comparisons between these data

and those from aircraft or revisions to the AIRS retrieval approach will modify this estimate of

the accuracy.

**8) Conclusion**

1        This paper describes the vertical resolution and error characteristics of retrievals of the

2        deuterium content (or the HDO/$H_2O$ ratio) of water vapor using AIRS radiances and then

3        evaluates the consistency between AIRS and TES retrievals of HDO and $H_2O$. We find that the

4        AIRS and TES deuterium content for the lower-troposphere (750 – 350 hPa) are consistent, or

5        within their calculated uncertainties, for the 5 year period in which TES observations span the

6        globe (2006-2010).  We find the total uncertainty for a single AIRS observation is ~30 per mil

7        with an accuracy of ~7.8 per mil.  These uncertainties can be compared to the observed  total

8        variability, which can range from approximately -350 to -50 per mil over the whole globe, as

9        observed by the Aura TES data (Worden *et al*. 2006) and shown in Figure 3 for AIRS data.

10        While only five days of comparisons are shown here for the purpose of evaluating the

11        retrieval approach and error characteristics of these AIRS retrievals, we expect to produce a

12        record of the AIRS-based deuterium content retrievals from the start of the mission (2002)

13        through the present. Because of computational limitations, we expect to process data from 45

14        degrees South to 65 degrees North at approximately four times the sampling of the Aura TES

15        measurements and with increased sampling (~3x) over the continental regions with the goal of

16        increasing this sampling once the initial record is completed and as additional resources become

17        available.

20        **Acknowledgements**

22        The research was carried out at the Jet Propulsion Laboratory, California Institute of
23        Technology, under a contract with the National Aeronautics and Space Administration.

24

26

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

Table 1: Comparison between averaged TES and AIRS HDO/$H_2O$ ratio (750-350 hPa). The units
are in parts per thousand relative to Standard Mean Ocean Water. The second column shows the
expected RMS based on the uncertainties of the TES and AIRS data. The third column shows the
actual RMS difference between TES and AIRS. The last column shows the mean difference.

| Date | Expected RMS (per mil / SMOW) | Actual RMS (per mil / SMOW) | Mean (TES-AIRS) (per mil / SMOW) |
|---|---|---|---|
| 2006-06-01 | 31.1 | 30.6 | -2.7 +/- 1.5 |
| 2007-06-02 | 30.0 | 31.9 | -0.6 +/- 1.5 |
| 2008-06-02 | 31.5 | 29.3 | 0.5 +/- 1.4 |
| 2009-07-06 | 31.6 | 27.1 | 0.7 +/- 1.4 |
| 2010-06-02 | 31.6 | 28.2 | 3.7 +/- 1.2 |

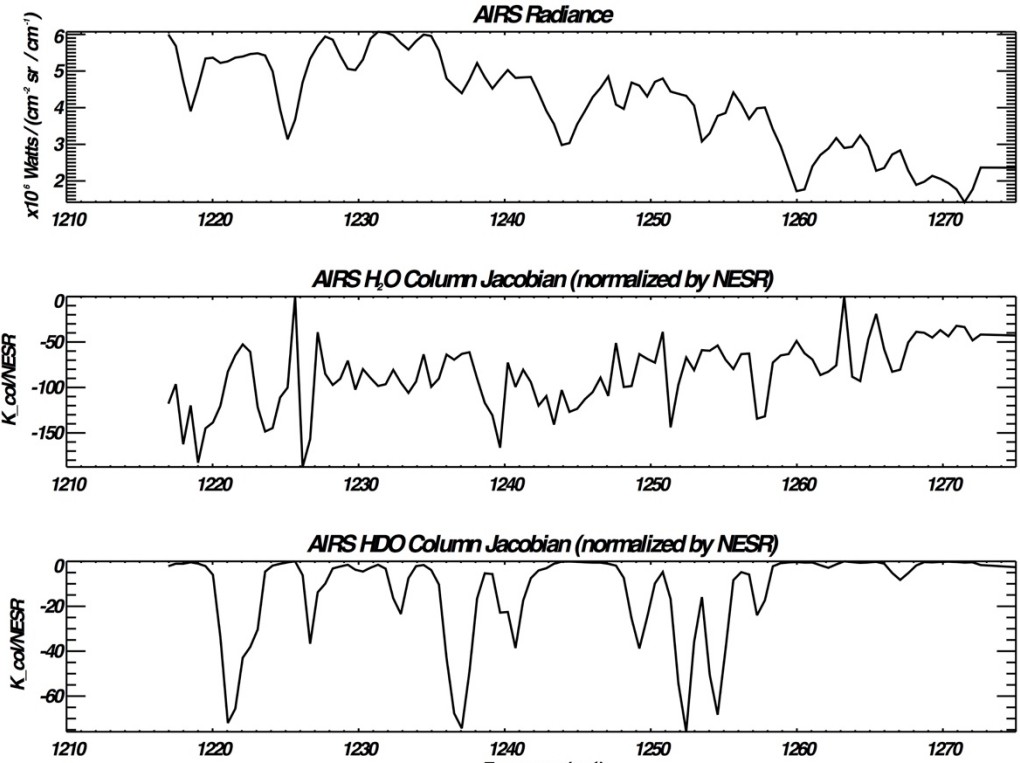

Figure 1: (top) AIRS radiance at approximately 8 microns for a typical tropical scene. (middle)
The total column (log) Jacobian for $H_2O$ normalized by the AIRS NESR. (bottom) The total
column (log) Jacobian for HDO normalized by the AIRS NESR.

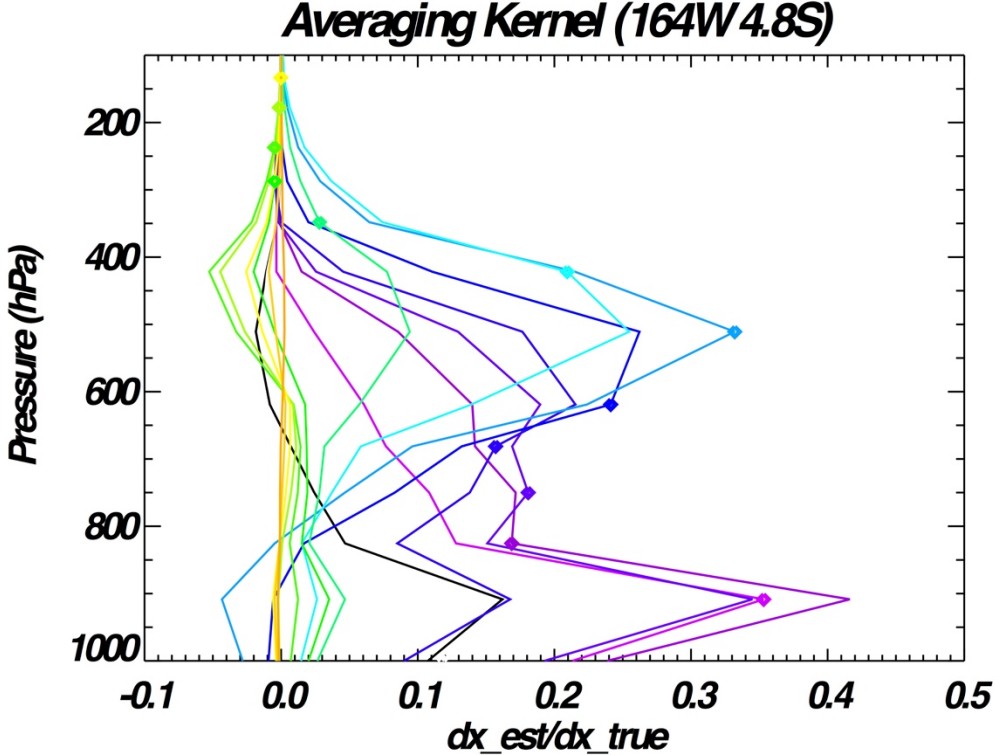

Figure 2: The rows of the averaging kernel matrix for the HDO retrieval corresponding to the
radiance shown in Figure 1. The different colors and symbols are used to indicate the pressure
levels corresponding to each row of the averaging kernel matrix.

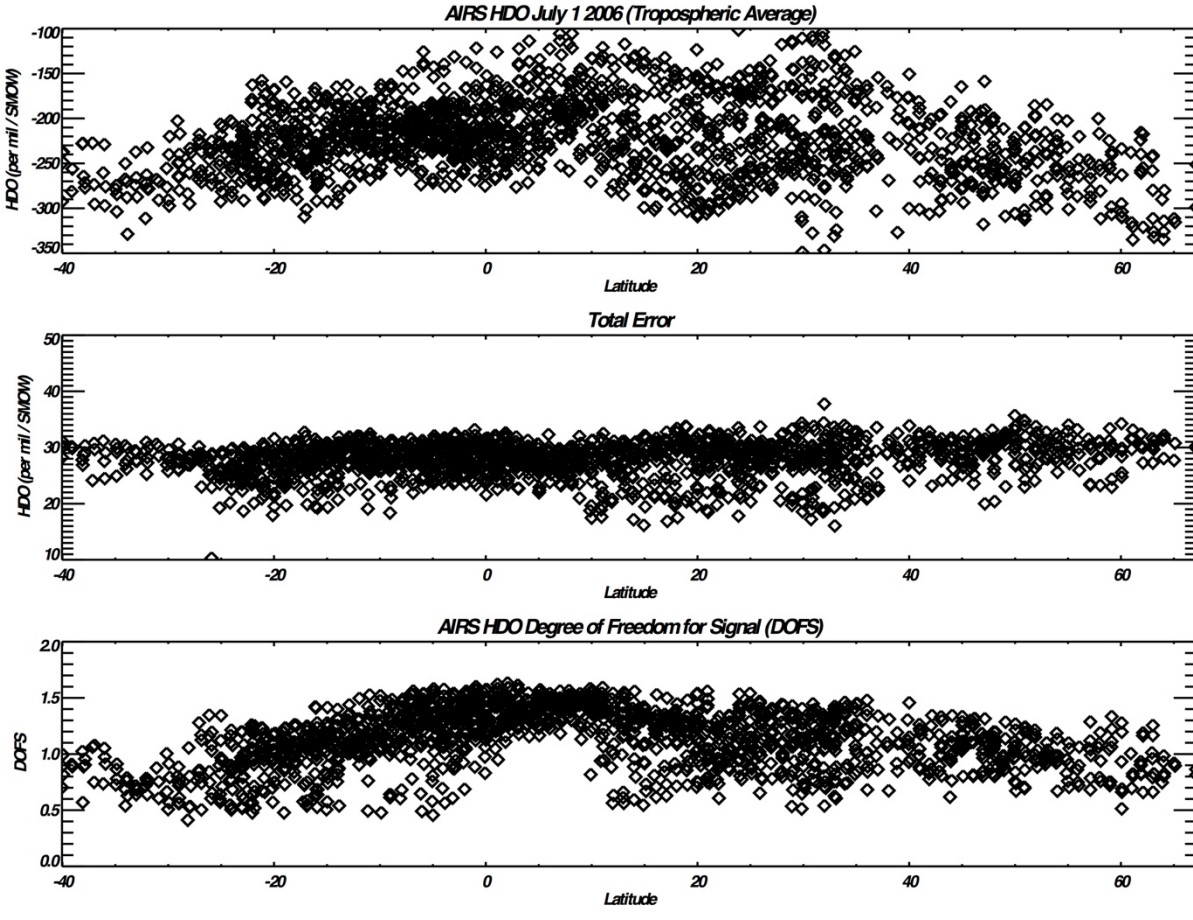

Figure 3: (top) The mean tropospheric deuterium content (in "per mil" or units of parts per
thousand relative to the deuterium content of the ocean or SMOW) for June 1 2006 as inferred
from AIRS radiance measurements. (middle) The total error for the measurements in the top
panel (also in units of per mil relative to SMOW). (bottom) The DOFS for the retrieval.

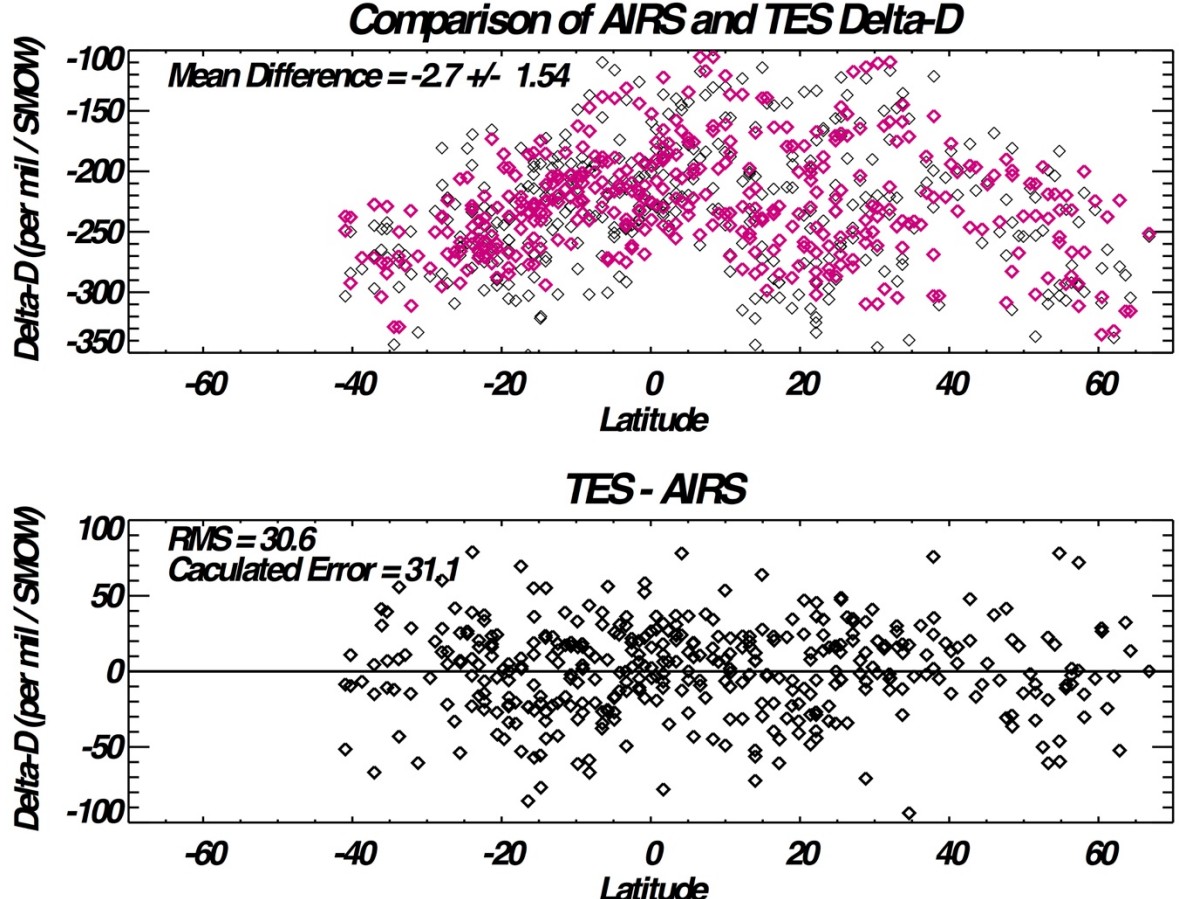

3 Figure 4: (top) Comparison of AIRS (red) and TES (black) delta-D for June 1 2006 (~600 co-
4 located observations). (bottom) The differences (after bias subtraction) between TES and AIRS
5 delta-D  measurements.

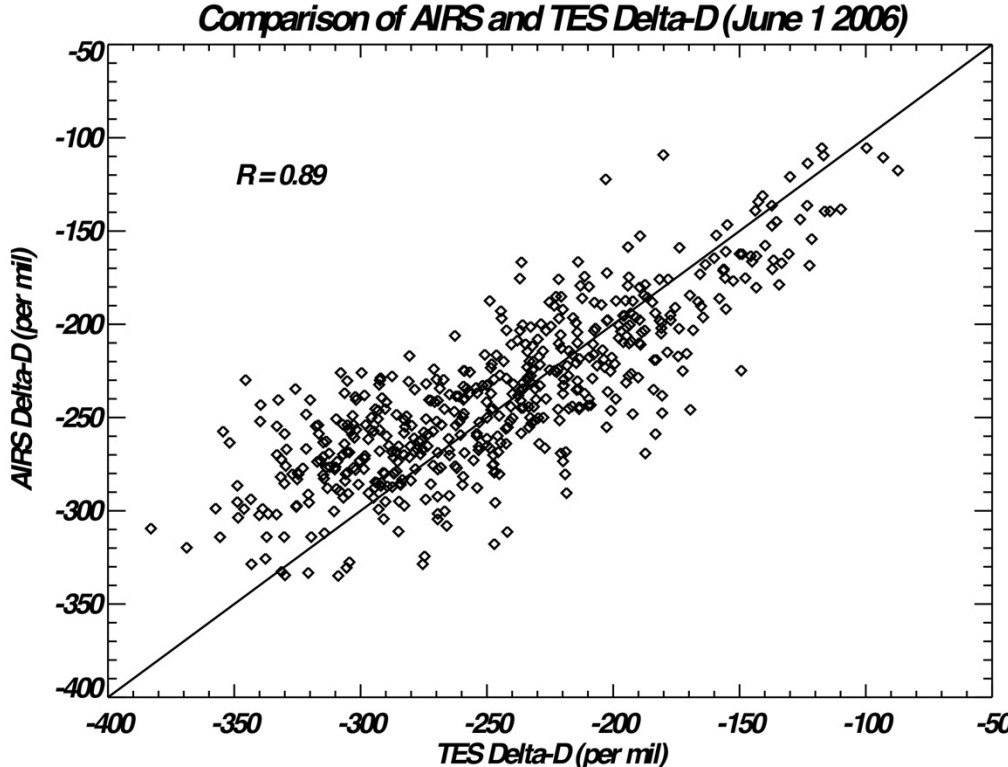

Figure 5: Comparison of the AIRS and TES deuterium content. The solid line is the one-to-one
line.

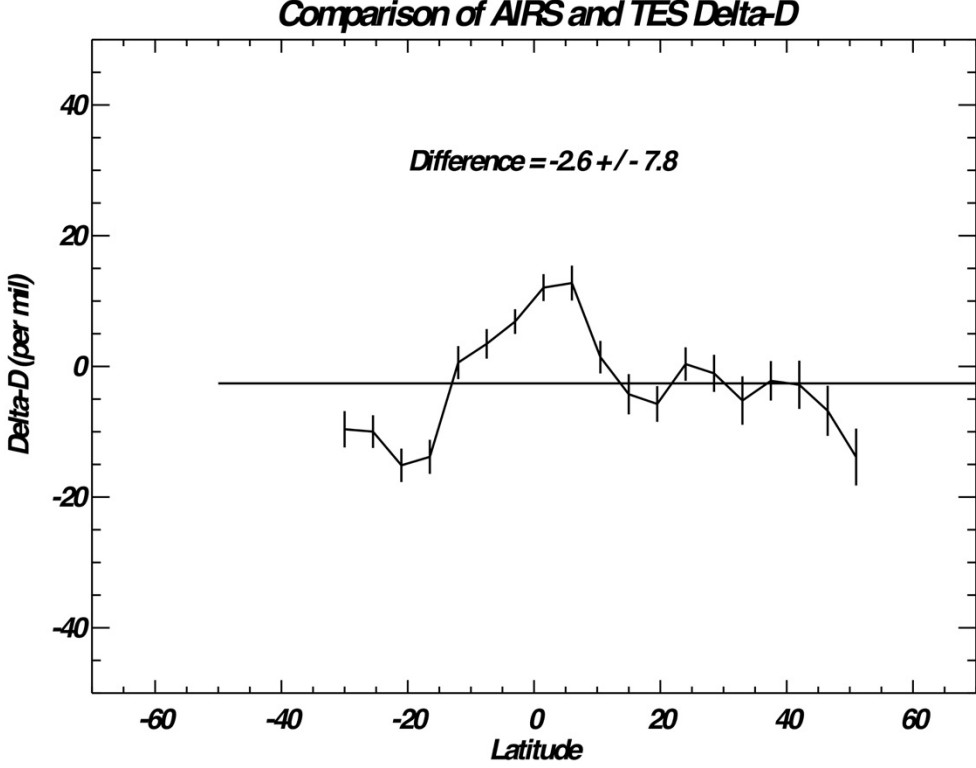

3 Figure 6: The Latitudinal differences between TES and AIRS Delta-D using co-located
4 observations for 5 days (approximately 600 observations per day) of data, spaced over 5
5 Northern Hemisphere summers between 2006 and 2010.
