# Peer review of "John R. Worden1, Susan S. Kulawik2, Dejian Fu1, Vivienne H. Payne1, Alan E. Lipton3, Igor"

_Atmospheric Measurement Techniques, 2018_

## Referee Comment (RC1) · Anonymous Referee #1 · 11 Jan 2019

This study presents the application of an existing retrieval methodology of HDO/H2O vertical profiles originally applied on TES, on AIRS thermal infrared measurements. The authors briefly remind the retrieval methodology, describe the error and sensitivity, and show a comparison with co-located TES retrievals. In my view, this is a welcome study as the capabilities of AIRS sensors for HDO/H2O ratio retrievals were unknown/not tested, and the sampling characteristics of AIRS offer great potential for isotopes related studies. The manuscript is short and generally convincing but the presentation is too minimalist and should be improved. Some discussions on previous improvements in characterizing HDO/H2O-H2O pairs retrieval is missing. I list a few comments which should be easily resolved by the authors.

[Figure]

**Specific comments**

- Introduction: A short introduction on water isotopes, their usefulness and a description on what are the remote sensing capabilities to observe HDO/H2O ratios in the free troposphere would be useful to strengthen the importance of this work and to smooth the feeling of reading a purely technical report.

- P2, Line 19: estimates of HDO/H2O ratios and not HDO

- P2, Line 20: Why only summertime TES global survey's? Do you mean boreal summertime?

- P2, Line 23: "We then compare the AIRS and TES data to evaluate and quantify the calculated uncertainties of the AIRS data" - To evaluate and quantify the calculated uncertainties sound a little odd. This needs to be rephrased.

- This paper is relatively short and yet there is a lot of statements about futures publications (P2, L17-18;P2, L23-24;P5, L29 – P6,L8). Some of them could be removed.

- P3, L8: There is a redundancy here of the statement that TES is part of the A-Train, it was just said in the previous sentence.

- P5, L9: "This retrieval algorithm can use radiances (..) to quantify and characterize geophysical observables appropriate for the corresponding radiance." – What is an appropriate geophysical observable? To retrieve different geophysical parameters?

- P5, L16-17: "in order to ensure that [the retrieval of] the ratio is optimized, as opposed (..)" [missing]

- P5, L29 – P6,L8: All this part describes the importance of including the 12 microns radiances for the methane retrieval. That is not interesting in the frame of this paper.

- P6, L17-19: Jacobians have not be defined. What does the -50 treshold represent? How is it calculated?

- P6, L22: "(..) partial derivative of the estimate relative to [partial derivative] of the true state". Or maybe in a language more accessible to potential users not familiar with optimal estimation: "the response of the retrieved state to perturbations of the true state"

- P6, L23: It is confusing to translate the example in terms of HDO/H2O ratios since the averaging kernels are for HDO.

- P6, L28-29: Schneider et al., 2012 proposed an a posteriori methodology to characterize the joint retrieval of H2O and HDO. The method allows to transform the products obtained in the log(H2O),log(HDO) space into a proxy state log(H2O),$\delta$D which is very useful for characterization. Moreover, the HDO/H2O ratio product is often used in pair with H2O it is therefore important to discuss the differences of sensitivity of H2O and HDO/H2O ratios. This is missing here.

- P7, L13-L15: There are a lot of measurements within the tropics with DOFS between 0.5 and 1 so I wouldn't generalize this situation to the whole tropics. This might be valid only for the averaging kernels shown.

- P8, L6->L11: All this part would better fit in the error characterization part

- Comparisons of AIRS and TES retrievals - In order to be really convincing, this part needs to be completed.

    – Would it be possible to show a scatter plot of AIRS versus TES?

- – What is the correlation between AIRS and TES retrievals?
- – Because this kind of product is used in pairs with humidity retrievals it is also interesting to show that both sounders show the same humidity-$\delta$D information and not only $\delta$D.
- – I didn't understand the error assessment reasoning. The mean bias across latitude is -2.6 permil, later on the authors assess the RMS to be 7.8 permil then the authors say the accuracy is 7.8 permil. Is this a mistake or do I miss something? The language between accuracy and precision should be clarified.
- – What about the latitudinal variations of the bias which are greater (-15 to 15 permil) than the mean standard error? It looks like there is a latitudinal bias, could it be caused by some dependence on temperature or humidity content?
- – Could you plot the data in Figure 5 until 40°S as in the previous figure?

- The conclusions could be more developed. One of the interest of this paper lies in the development of a HDO retrieval methodology from AIRS data which was unknown and opens great perspectives for users interested in such measurements. In this context, a word on the future plans of the authors on processing more AIRS data, or not, would be interesting.

- P9, L8: Please reference the natural variability of $\delta$D

**Technical corrections**

- Abstract, L17: Northern instead of N;

- P1, L28: a verb is missing

- L29, degrees

- P4 , L30 : Description of Retrieval Approach -> Description of the retrieval approach

- P5, L29 : (e.g. Figures 1-4).

- P7, L4: add degrees to latitude

- P7, L8: use the delta Greek notation $\delta$

- Figure 4: A legend is missing, what is TES and what is AIRS?
* * *

---

## Referee Comment (RC2) · Lars Hoffmann (Referee) · 19 Feb 2019

Dear authors,

the second review of your paper could not be provided it time. I am submitting this additional review so that the open discussion of your manuscript can be closed.

Best regards

Lars Hoffmann

General comments

[Figure]

As this study is targeting the preparation of a new Earth Science Data Record covering AIRS HDO/H2O observations, it clearly has a high scientific significance. The manuscript itself is clear and concise, but I would agree with Reviewer #1 that the presentation is indeed somewhat "minimalistic" and could be extended and improved. Please carefully follow suggestions and comments provided by Reviewer #1 and those listed below so that the paper can be published soon.

Specific comments

p3, l4: The AIRS swath width is 1650 km (Aumann et al., 2003) rather than 1250 km.

p5, l25-26: Although the AIRS noise is characterized well for individual channels, in other work I noticed noise can be spectrally correlated between neighboring channels, which is due to the 1-D linear detector arrays of AIRS sharing the same electric module (Pagano et al., 2008). This may be too specific to discuss in your paper; I just wondered if you considered this?

Pagano, T. S., Aumann, H. H., Schindler, R., Elliott, D., Broberg, S., Overoye, K., and Weiler, M. H.: Absolute radiometric calibration accuracy of the Atmospheric Infrared Sounder (AIRS), in: Proc. SPIE, vol. 7081, doi:10.1117/12.795445, 2008.

p7, l20-29: Are the HDO retrieval results correlated with the simultaneous H2O retrievals? Does the AVK matrix show any correlations between these retrieval variables?

Fig. 2: Maybe show also the integral of the AVKs, to indicate the amount of measurement information in the retrieval results? (Fix "are _used_ to indicate" in the caption.)

Fig. 4: A legend/definition of the colors used for the plot on the top seems to be missing.

Technical corrections

p1, l17: N. -> Northern (also in other places)

p1, l22: ...reduced spectral resolution _of AIRS_ (for clarity?)

[Figure]

p1, l24: Suggest to remove reference (Worden et al., 2004) from abstract.

p1, l27-28: Please fix incomplete sentence.

p1, l29: Add degree symbols to "30 S and 50 N" (also in other places)?

p2, l2: The copyright statement "All rights reserved." is not allowed in the given form, I think, please see https://www.atmospheric-measurement-techniques.net/about/licence_and_copyright.html for details.

p2, l4: "Introduction:" -> "Introduction"

p3, l11: PAN, -> PAN__

p3, l22: Earth Science Data Records (ESDR's) -> ESDR's (acronym was already introduced)

p4, l25: "a version of the v4 AIRS" -> "version 4 of the AIRS" (?)

p5, l6: will _only briefly_ summarize (?)

p5, l9-10: Not sure if "...appropriate for the corresponding radiance." is a good phrase here?

p6, l4-6: Remove redundant sentence.

p6, l31: Change date format to "1 July 2016" (also in other places).

p7, l2: we _can_ only (?)

p8, l14: indicate_s_
* * *

---

## Author Comment (AC1) · 3 Mar 2019

I would like to greatly thank both reviewers for their detailed review and comments. Reviewing and editing these papers is quite a bit of (effectively voluntary) work and both reviews really went into great detail on fixing the presentation. With respect to the comment on "minimalism", I actually was striving for minimalism in this paper but apparently overshot my goal! ☺. For example, I did not want to write (yet) another paper full of the optimal estimation description in all of its gory, (or glory?) equation detail but instead report on the basic notions….. that the AIRS radiances can be used to generation global deuterium content retrievals and that their vertical resolution is about the same as TES but with slightly poorer uncertainties, and that we will soon produce a long record of the data that spans most of the globe and can hopefully produce scientific awesomeness.

Again, thanks for the review. Below are the comments and my responses.

**Response to Reviewer 1**

General Comment: This study presents the application of an existing retrieval methodology of HDO/H2O vertical profiles originally applied on TES, on AIRS thermal infrared measurements. The authors briefly remind the retrieval methodology, describe the error and sensitivity, and show a comparison with co-located TES retrievals. In my view, this is a welcome study as the capabilities of AIRS sensors for HDO/H2O ratio retrievals were unknown/not tested, and the sampling characteristics of AIRS offer great potential for isotopes related studies. The manuscript is short and generally convincing but the presentation is too minimalist and should be improved. Some discussions on previous improvements in characterizing HDO/H2O-H2O pairs retrieval is missing. I list a few comments which should be easily resolved by the authors.

**Specific Comments:**

- Introduction: A short introduction on water isotopes, their usefulness and a description on what are the remote sensing capabilities to observe HDO/H2O ratios in the free troposphere would be useful to strengthen the importance of this work and to smooth the feeling of reading a purely technical report.

  **Response**: I added a paragraph at the front that describes a bit of history on water isotope measurements, and how these vapor based measurements have helped address global water / carbon questions.

- P2, Line 19: estimates of HDO/H2O ratios and not HDO

  **Response**: Added "and their ratio". We actually do retrieve HDO and $H_2O$ seaparately even if the retrieval setup optimizes the ratio.

- P2, Line 20: Why only summertime TES global survey's? Do you mean boreal summertime?

  **Response**: added boreal and added a statement about current limited processing capabilities

- P2, Line 23: "We then compare the AIRS and TES data to evaluate and quantify the calculated uncertainties of the AIRS data" - To evaluate and quantify the calculated uncertainties sound a little odd. This needs to be rephrased.

  **Response**: removed "and quantify"

- This paper is relatively short and yet there is a lot of statements about futures publications (P2, L17-18;P2, L23-24;P5, L29 – P6,L8). Some of them could be removed.

  **Response**: Removed most of these references where appropriate and modified some of the language about the utility of the 12 micron band for constraining atmospheric temperature.

- P3, L8: There is a redundancy here of the statement that TES is part of the A-Train, it was just said in the previous sentence.

  **Response** removed

- P5, L9: "This retrieval algorithm can use radiances (..) to quantify and characterize geophysical observables appropriate for the corresponding radiance." – What is an appropriate geophysical observable? To retrieve different geophysical parameters?

  **Response**: changed "appropriate" to "that affect". I think this wording is appropriate but terse. I can also add another line such as (e.g. the ozone concentrations affect radiances in the 9.6 micron ozone band) but that seems too wordy.

- P5, L16-17: "in order to ensure that [the retrieval of] the ratio is optimized, as opposed (..)" [missing]

  **Response** fixed

- P5, L29 – P6,L8: All this part describes the importance of including the 12 microns radiances for the methane retrieval. That is not interesting in the frame of this paper.

  **Response** (fixed in above response, hopefully ☺ ).

- P6, L17-19: Jacobians have not be defined. What does the -50 treshold represent? How is it calculated?

  **Response:**  I added the definition for a Jacobian and changed the language around.. basically 2% is equivalent to 1/50... 1% = 1/100 etc.

- P6, L22: "(..) partial derivative of the estimate relative to [partial derivative] of the true state". Or maybe in a language more accessible to potential users not familiar with optimal estimation: "the response of the retrieved state to perturbations of the true state"

  **Response**: changed

- P6, L23: It is confusing to translate the example in terms of HDO/H2O ratios since the averaging kernels are for HDO.

  **Response** thanks for pointing this one out.. I adjusted the language accordingly and added a sentence about how the information about HDO/H$_2$O is limited by HDO; I also added a reference.

- P6, L28-29: Schneider et al., 2012 proposed an a posteriori methodology to characterize the joint retrieval of H2O and HDO. The method allows to transform the products obtained in the log(H2O),log(HDO) space into a proxy state log(H2O),δD which is very useful for characterization. Moreover, the HDO/H2O ratio product is often used in pair with H2O it is therefore important to discuss the differences of sensitivity of H2O and HDO/H2O ratios. This is missing here.

  **Response** I added language on how the averaging kernels for H$_2$O are different than that for HDO and that Schneider et al discusses an approach to use these data with simple models while accounting for the different sensitivities.

- P7, L13-L15: There are a lot of measurements within the tropics with DOFS between 0.5 and 1 so I wouldn't generalize this situation to the whole tropics. This might be valid only for the averaging kernels shown.

  **Response**: adjusted the language .. now we say "many" which implies "not all"

- P8, L6->L11: All this part would better fit in the error characterization part

  **Response**: moved

- Comparisons of AIRS and TES retrievals - In order to be really convincing, this part needs to be completed.

  – Would it be possible to show a scatter plot of AIRS versus TES?

- – What is the correlation between AIRS and TES retrievals?

  **Response**: I am not that convinced this is a meaningful figure as it is the difference between AIRS and TES that can be used to determine if the AIRS data is (relatively) well characterized. Having said that I have included it here in case readers find it useful. The correlation for this day is 0.89.

- – Because this kind of product is used in pairs with humidity retrievals it is also interesting to show that both sounders show the same humidity-δD information and not only δD.

  **Response**: I dont think this comparison is of use to this specific paper as it shows that the pairs generated by AIRS are similar to those from TES but slightly different, as expected because the sensitivity and errors are different. For this reason I would prefer not to show here, but will show in subsequent papers when we start looking at the science!

- – I didn't understand the error assessment reasoning. The mean bias across latitude is -2.6 permil, later on the authors assess the RMS to be 7.8 permil then the authors say the accuracy is 7.8 permil. Is this a mistake or do I miss something? The language between accuracy and precision should be clarified.
- – What about the latitudinal variations of the bias which are greater (-15 to 15 permil) than the mean standard error? It looks like there is a latitudinal bias, could it be caused by some dependence on temperature or humidity content?

    **Response:** I attempted to clean up the language here, hopefully it's a bit more clear! I also added language that the latitudinal variations are typically due to uncertainties in temperature, water vapor, and spectroscopy, as well as differences in the vertical resolution.

- – Could you plot the data in Figure 5 until 40°S as in the previous figure?

    **Response:** Done!

- The conclusions could be more developed. One of the interest of this paper lies in the development of a HDO retrieval methodology from AIRS data which was unknown and opens great perspectives for users interested in such measurements. In this context, a word on the future plans of the authors on processing more AIRS data, or not, would be interesting.

    Response: Added a paragraph on current and future plans with respect to building an ESDR.

- P9, L8: Please reference the natural variability of $\delta D$

    Response: Added statement about Figure 3 and cited a TES paper.

    **Technical corrections**

    • Abstract, L17: Northern instead of N; • P1, L28: a verb is missing  (fixed)

    - L29, degrees  (fixed)

- P4 , L30 : Description of Retrieval Approach -> Description of the retrieval approach (fixed)
- P5, L29 : (e.g. Figures 1-4). (fixed)
- P7, L4: add degrees to latitude  (fixed)
- P7, L8: use the delta Greek notation $\delta$ (this is stylistic, I have added "or $\delta$" instead)
- Figure 4: A legend is missing, what is TES and what is AIRS? (fixed)

**General Comment**

As this study is targeting the preparation of a new Earth Science Data Record covering AIRS HDO/H2O observations, it clearly has a high scientific significance. The manuscript itself is clear and concise, but I would agree with Reviewer #1 that the presentation is indeed somewhat "minimalistic" and could be extended and improved. Please carefully follow suggestions and comments provided by Reviewer #1 and those listed below so that the paper can be published soon.

**Specific comments**

- p3, l4: The AIRS swath width is 1650 km (Aumann et al., 2003) rather than 1250 km.

Response: (fixed)

- p5, l25-26: Although the AIRS noise is characterized well for individual channels, in other work I noticed noise can be spectrally correlated between neighboring channels, which is due to the 1-D linear detector arrays of AIRS sharing the same electric module (Pagano et al., 2008). This may be too specific to discuss in your paper; I just wondered if you considered this?

Response: We have not explicitly addressed this issue and in fact the same is true with the TES data because the data are apodized but the calculated errors assume the noise is random (or not apodized). On the other hand the apodization is accounted for in the TES retrievals when we calculate the forward model radiance. In contrast with the TES data where we can account for the apodiziation it is not clear how to account for these noise correlations in the AIRS data. Instead we added a statement that the noise is assumed to be random as we are unable to account for correlations between channels. The effect of this assumption is that the calculated errors will be too large and is therefore a conservative estimate of the uncertainties.

Pagano, T. S., Aumann, H. H., Schindler, R., Elliott, D., Broberg, S., Overoye, K., and Weiler, M. H.: Absolute radiometric calibration accuracy of the Atmospheric Infrared Sounder (AIRS), in: Proc. SPIE, vol. 7081, doi:10.1117/12.795445, 2008.

- p7, l20-29: Are the HDO retrieval results correlated with the simultaneous H2O re-trievals? Does the AVK matrix show any correlations between these retrieval variables?

Response: Yes! These correlations are addressed 1) in model comparisons by applying the averaging kernels for both HDO and $H_2O$ to the model (e.g. Risi *et al*. 2013) or 2) by calculating the resulting error (e.g. Worden *et al*. 2006) and including that in the error budget, or mitigating further by 3) projecting to HDO-$H_2O$ pairs as discussed by the first reviewer. I have discussed the pairing approach based on the first reviewers comments and will add the Risi reference as well.

Fig. 2: Maybe show also the integral of the AVKs, to indicate the amount of measurement information in the retrieval results? (Fix "are _used_ to indicate" in the caption.)

Response: Fixed caption. The integral of the AVKs (or trace actually) is shown in the bottom panel of Figure 3 and discussed in the text.

Fig. 4: A legend/definition of the colors used for the plot on the top seems to be missing.

Response: Fixed!

Technical corrections
p1, l17: N. -> Northern (also in other places) (fixed)
p1, l22: ...reduced spectral resolution _of AIRS_ (for clarity?) (removed sentence as it was confusing to have in the abstract)

p1, l24: Suggest to remove reference (Worden et al., 2004) from abstract. p1, l27-28: Please fix incomplete sentence. (see above)

p1, l29: Add degree symbols to "30 S and 50 N" (also in other places)? I have added the word "degrees" instead.

p2, l2: The copyright statement "All rights reserved." is not allowed in the given form, I think, please see https://www.atmospheric-measurement- techniques.net/about/licence_and_copyright.html for details.

Response: I have to use this copyright for JPL during the submission phase. Once / if the paper is accepted I put in another form where JPL puts in a modification to the Copernicus agreement.

p2, l4: "Introduction:" -> "Introduction" p3, l11: PAN, -> PAN__

fixed

p3, l22: Earth Science Data Records (ESDR's) -> ESDR's (acronym was already introduced)

fixed

p4, l25: "a version of the v4 AIRS" -> "version 4 of the AIRS" (?) (fixed)

 p5, l6: will _only briefly_ summarize (?)  (fixed)

p5, l9-10: Not sure if "...appropriate for the corresponding radiance." is a good phrase here?

(changed, see comment from reviewer 1)

p6, l4-6: Remove redundant sentence. (fixed)

p6, l31: Change date format to "1 July 2016" (also in other places); I think this is a US Versus Europe date thing ☺.  Can I keep as is? Its like asking me to drive in the left lane ☺.

p7, l2: we _can_ only (?) (fixed)

p8, l14: indicate_s_ (fixed)

---

## Author Comment (AC2) · 3 Mar 2019

See supplemental text. I have placed both reviews in the supplemental for ease of cross-comparison

Please also note the supplement to this comment:
https://www.atmos-meas-tech-discuss.net/amt-2018-372/amt-2018-372-AC2-supplement.pdf